# Reduction of coastal lighting decreases seabird strandings

**Tori V. Burt**[1]*, **Sydney M. Collins**[1], **Sherry Green**[2], **Parker B. Doiron**[1], **Sabina I. Wilhelm**[3], **William A. Montevecchi**[4]

**1** Departments of Psychology and Biology, Memorial University of Newfoundland and Labrador, St. John's, Newfoundland and Labrador, Canada, **2** Quinlan Brothers Ltd., Bay de Verde, Newfoundland and Labrador, Canada, **3** Canadian Wildlife Service, Environment and Climate Change Canada, St. John's, Newfoundland and Labrador, Canada, **4** Departments of Psychology, Biology, and Ocean Sciences, Memorial University of Newfoundland and Labrador, St. John's, Newfoundland and Labrador, Canada

\* tburt@mun.ca

**Data Availability Statement:** All data files pertaining to Leach's Storm-Petrel strandings are available on the repository figshare (Code/Software is now available at: https://doi.org/10.6084/m9.figshare.25229471; data files are now available at:

## Abstract

Artificial light at night (ALAN) is negatively impacting numerous species of nocturnally active birds. Nocturnal positive phototaxis, the movement towards ALAN, is exhibited by many marine birds and can result in stranding on land. Seabird species facing major population declines may be most at risk. Leach's Storm-Petrels (*Hydrobates leucorhous*) are small, threatened seabirds with an extensive breeding range in the North Atlantic and North Pacific Oceans. The Atlantic population, which represents approximately 40–48% of the global population, is declining sharply. Nocturnal positive phototaxis is considered to be a key contributing factor. The Leach's Storm-Petrel is the seabird species most often found stranded around ALAN in the North Atlantic, though there is little experimental evidence that reduction of ALAN decreases the occurrence of stranded storm-petrels. During a two-year study at a large, brightly illuminated seafood processing plant adjacent to the Leach's Storm-Petrel's largest colony, we compared the number of birds that stranded when the lights at the plant were on versus significantly reduced. We recorded survival, counted carcasses of adults and juveniles, and released any rescued individuals. Daily morning surveys revealed that reducing ALAN decreased strandings by 57.11% (95% CI: 39.29% - 83.01%) per night and night surveys revealed that reducing ALAN decreased stranding of adult birds by 11.94% (95% CI: 3.47% - 41.13%) per night. The peak stranding period occurred from 25 September to 28 October, and 94.9% of the birds found during this period were fledglings. These results provide evidence to support the implementation of widespread reduction and modification of coastal artificial light as a conservation strategy, especially during avian fledging and migration periods.

## Introduction

Many seabird populations have been negatively impacted by artificial light at night (ALAN) [1, 2]. Light from coastal towns, lighthouses, fishing vessels, and offshore hydrocarbon platforms

https://doi.org/10.6084/m9.figshare.25229414; read me file is now available at: https://doi.org/10.6084/m9.figshare.25229528), moon illumination data are available at Time and Date AS (https://www.timeanddate.com/), and wind speed and direction data are available at Environment and Climate Change Canada (https://climate.weather.gc.ca/climate_data/daily_data_e.html?hlyRange=1994-02-01%7C2023-11-01&dlyRange=2002-01-01%7C2023-11-01&mlyRange=2004-12-01%7C2004-12-01&StationID=10818&Prov=NL&urlExtension=_e.html&searchType=stnName&optLimit=yearRange&StartYear=2021&EndYear=2022&selRowPerPage=25&Line=0&searchMethod=contains&Month=11&Day=5&txtStationName=Grates+Cove&timeframe=2&Year=2021). The weather data underlying the results presented in the study are available from CustomWeather, Inc. (https://customweather.com/) (email: sales@customweather.com). These weather data are owned by CustomWeather, Inc and were used with permission from the company.

**Funding:** Funding was provided by Environment and Climate Change Canada (https://www.canada.ca/en/environment-climate-change.html) to W.A. M. (no. GCXE22C307) and the Natural Sciences and Engineering Research Council of Canada by a Discovery Grant (https://www.nserc-crsng.gc.ca/index_eng.asp) to W.A.M. (no. 006872) and an Undergraduate Student Research Award to T.V.B. These funders did not play any role in the study design, data collection and analysis, decision to publish, or preparation of the manuscript.

**Competing interests:** The authors have declared that no competing interests exist.

attract marine birds, often causing them to collide with man-made structures, resulting in injury, oiling, and stranding [1, 2]. Stranded seabirds have difficulty taking off from land due to injury, stress, and disorientation, resulting in some seeking refuge in dark and inaccessible locations [3]. Without intervention from rescue groups, stranded birds are subjected to predation, starvation, and dehydration, and mortality is probable [1, 3, 4].

Little is known about why seabirds cluster around light, but several non-mutually exclusive and potentially interactive hypotheses have been proposed. First, birds may confuse artificial lights with bioluminescence [5, 6]. Some seabirds feed on bioluminescent organisms [5–7], and it is plausible that birds are not able to differentiate between natural and artificial light [5]. Second, ALAN may interfere with the ability of marine birds to navigate using light from the moon and stars or their ability to sense magnetic orientation [5, 8]. Third, ALAN may disorient individuals, causing them to be "caught" in a light catch basin from which they have difficulty escaping [1]. Light effects may be intensified in fledglings, as their retinal development is not completed until after fledging [9].

The extent to which seabird strandings are caused by a phototactic response, and how that response interacts with other environmental factors, is also unclear. For example, seabirds may be pushed onshore by gale-force winds and become stranded [10, 11]. In addition, more birds strand around sources of ALAN when cloud cover is high [5] and moon illumination is low [4, 10, 12]. Studies have shown distinct temporal peaks in stranding that associate with the affected species' fledging period and migration [4, 10, 13, 14].

Of the seabird orders, Procellariiformes is among the most vulnerable to the effects of ALAN [15]. The Leach's Storm-Petrel (*Hydrobates leucorhous*) is one of the most nocturnally active procellariforms [16] and is particularly vulnerable to the effects of ALAN in the North Atlantic [4, 10, 13]. During the past 40 years, the Northwest Atlantic population of Leach's Storm-Petrels has declined by more than 50%, and as a result, the species is listed as 'Vulnerable' by the International Union for the Conservation of Nature (IUCN) and 'Threatened' by the Committee on the Status of Endangered Wildlife in Canada (COSEWIC) and the Government of Newfoundland and Labrador [17–19]. This population decline is likely attributable to numerous factors, including positive phototaxis [18, 20].

It has been widely suggested that turning off the lights at night should be used as a conservation strategy to decrease storm-petrel strandings [10, 21]. Efforts to reduce strandings at common stranding sites by reducing ALAN have been successful with other storm-petrel and procellarid species [5, 22, 23]. To our knowledge, only one study has attempted to experimentally determine the effect of ALAN on the occurrence of stranded Leach's Storm-Petrels [12]. This study, however, had very small numbers of stranded birds, and light conditions only differed between years [12], so annual differences could have accounted for the perceived effects of ALAN.

Our research investigated intra- and inter-annual variation in the number of strandings at a major light source near the Leach's Storm-Petrels' largest colony on Baccalieu Island, Newfoundland and Labrador, Canada, throughout the breeding period (May - November) of 2021 and 2022. We hypothesized that: 1) the stranding of storm-petrels at coastal industrial sites is a result of positive phototaxis [10, 13, 20, 24], and predicted that significantly reducing ALAN by turning off 15 out of 27 lights at an industrial site would reduce the number of stranded birds; 2) environmental conditions influence light visibility and subsequently the number of storm-petrels stranding around ALAN [5, 10–12, 25] and predicted that more birds would strand when wind speed and cloud cover are high, the wind is blowing onshore, and moon illumination is low; 3) fledglings are more susceptible to the effects of ALAN than adults [10, 13], and predicted that fledglings would be disproportionately represented in the stranded

birds recovered and that the peak stranding period would occur during the Leach's Storm-Petrel fledging period from early September through October [10, 13].

## Methods

### Study site

Data were collected at the Quinlan Brothers Ltd. seafood processing plant in Bay de Verde, Newfoundland and Labrador, Canada (Fig 1A; 48.09780, -52.89831). Bay de Verde is within 10 km of Baccalieu Island (48.15089, -52.79700), the site of the Leach's Storm-Petrels' largest colony, estimated at 2 million pairs [26] (Fig 1B). Depending on weather conditions, skyward lighting from Bay de Verde can be seen from Baccalieu Island, and citizen reports indicated that thousands of storm-petrels were stranding at the plant each year, making it an ideal location to conduct this experiment. The building is 6,919 m² [27] and has 16 bright overhanging light-emitting diodes (LEDs) along the top of the building in front of the wharf facing northwest into the harbour and Conception Bay (Figs 2 and 3). There are an additional 22 lights on the back and sides of the plant, and nine lights on the storage building and parking lot area (Fig 3). Lights are either overhanging or attached to the wall of the building, those attached to the wall are dimmer relative to the overhanging LEDs, meaning the brightest lighting is on the wharf-side of the plant, where most birds are attracted. There are also LEDs and high-pressure sodium lights along the wharf and around the vicinity of the plant.

### Lighting regime

Starting June 25 2021 and ending November 10 2021, 12 of the 16 overhanging LEDs on the front of the building were turned on and off according to a 28-day random schedule which aligned with the lunar cycle, for a total of 14 days of each treatment in each 28-day period (Fig 2). This period was chosen because reduced numbers of stranded storm-petrels have been

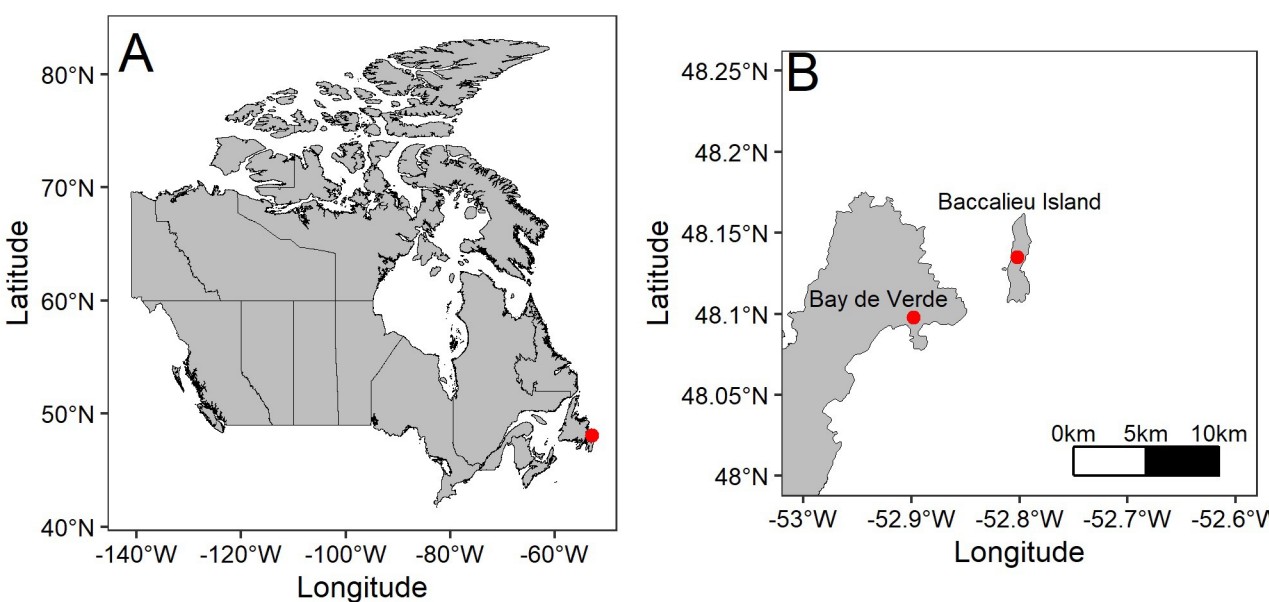

**Fig 1.** Map of Canada (A) with a red point indicating the study site Bay de Verde, Newfoundland and Labrador and (B) the study site relative to the world's largest Leach's Storm-Petrel colony on Baccalieu Island, Newfoundland and Labrador indicated with red points.

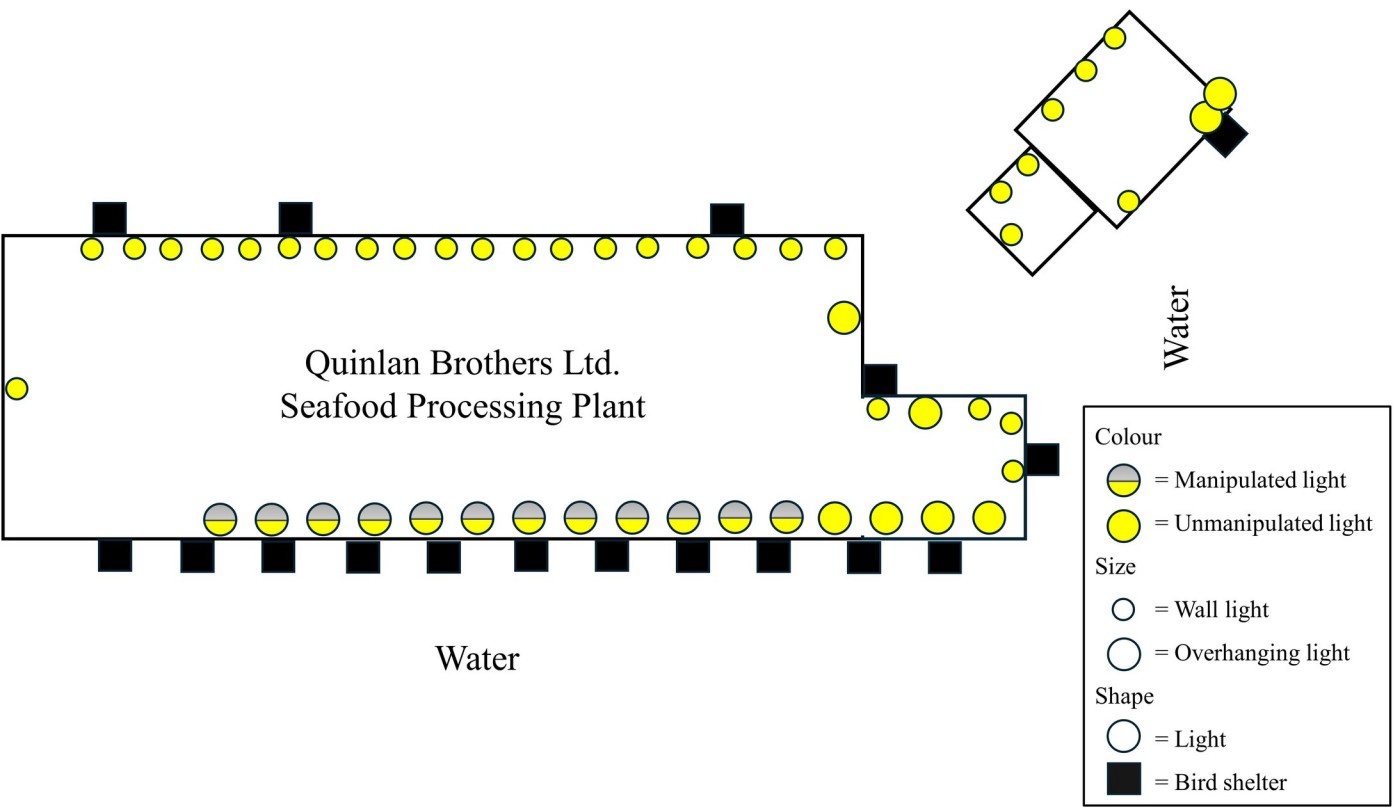

**Fig 2. Top-down diagram of the Quinlan Brothers Ltd. seafood processing plant in 2022 (not to scale).** The yellow and grey circles indicate the approximate locations of the LEDs on the front of the building that were manipulated (turned on and off) during the experiment and the yellow circles indicate the approximate location of the unmanipulated LEDs that remained on during the entire study period. Smaller circles represent wall lights and larger circles represent overhanging lights. The black squares indicate the approximate locations of the bird shelters. The length of the front of the building is approximately 200 metres long and 10 metres high.

associated with a fuller moon [10]. This schedule proved difficult for the plant staff to follow, so in 2022, we implemented a schedule with which the lights were turned on during odd days of the month and off during even days. This began on May 1 2022 and ended on November 10 2022. The schedule was followed as closely as possible when the plant was in operation but sometimes deviated (S2 Fig). Lights at the back and sides of the plant were on at all times and lights at the front were turned on briefly, regardless of the lighting schedule, when vessels were unloading catch for safety purposes.

## Data collection

**Daily morning surveys.** Leach's Storm-Petrels are nocturnally active burrow-nesters that seek refuge in dark confined spaces when stranded [3]. In 2019, we incidentally observed Leach's Storm-Petrels finding refuge in bait boxes used by pest control companies to control rat populations at two industrial sites where stranding events were being investigated (this site in Bay de Verde and the Holyrood Thermal Generation Station; [7]). As a result, we used these same boxes as bird shelters (29 x 25 x 15 cm Protecta EVO Express Bait Station after removing the inner contents) deployed around the perimeter of the building to safely capture and protect stranded birds from predation or injury. Fourteen boxes were deployed in 2021, and an additional three (17 total) were deployed in 2022 (Fig 2). Bird shelters were checked each morning from April 28 to November 30, 2021 and from May 3 to November 10, 2022. Surrounding

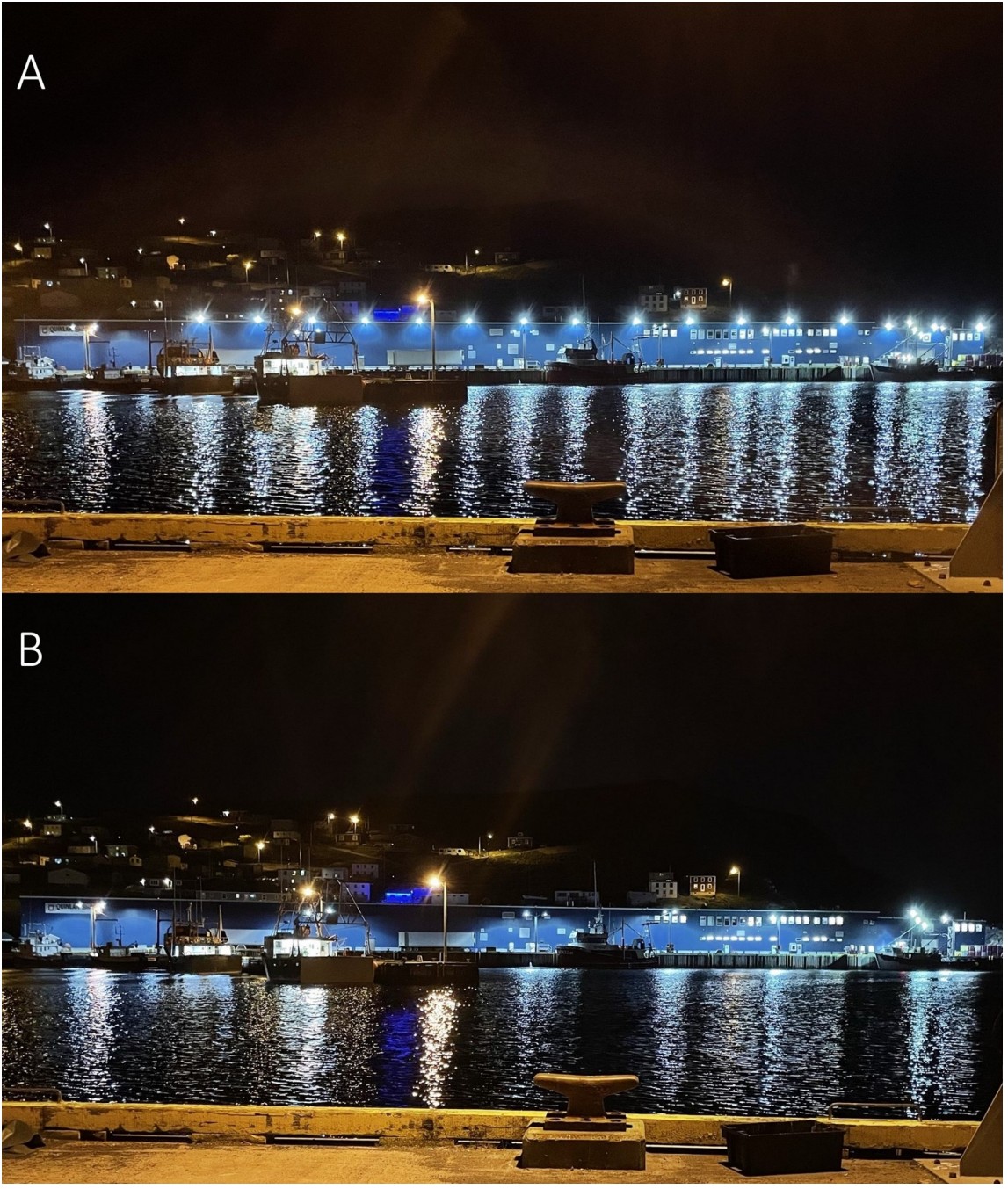

**Fig 3.** Photos taken in 2021 of the Quinlan Brothers Ltd. seafood processing plant with the LEDs on the front of the building on (A) and with nearly all of them off (B) (Photos by S.M. Collins).

areas (e.g. under concrete blocks or stairs) including the back and sides of the plant, and the storage building were also inspected, and all storm-petrels found were collected. Morning checks were sometimes not completed when the plant was closed, or when nightly surveys (see below) were performed. Healthy birds were released the following evening at dusk unless poor weather conditions made it unsafe to release them. In this case, birds were kept for an additional night to ensure safe release. Injured birds were kept overnight and released the following

evening if they recovered. Each morning, we recorded the date, lighting condition (on/ reduced), fog conditions, and number of live and dead birds. There were 158 surveys conducted in 2021 and 156 surveys conducted in 2022 (see S1 Fig for a breakdown of surveys by month). Due to missing light condition information, ($n_{lightson}$ = 111, $n_{lightsreduced}$ = 174, not available = 29) we removed 29 nights from our analysis. The lights at the plant remained off for much of September and October in 2021 and 2022, despite the lighting schedule, because the plant was not processing fish.

**Night surveys.**   Morning checks do not capture all birds that strand due to the capacity and variable catch of bird shelters, predators, and birds moving to inaccessible locations. To supplement the morning collections, 13 nightly surveys were performed opportunistically from June 25 to October 11, 2021 ($n_{lightson}$ 2021 = 3, $n_{lightsreduced}$ 2021 = 8) and 22 nightly surveys were performed approximately bi-weekly from April 20 to October 21, 2022 ($n_{lightson}$ 2022 = 8, $n_{lightsreduced}$ 2022 = 14; S1 Fig). Due to missing data, two of these had to be removed from our analysis ($n_{lightson}$ = 11, $n_{lightsreduced}$ = 22, not available = 2). Starting around 22:00, we patrolled the plant perimeter every 30–45 minutes until around 04:00 depending on the level of bird activity. We collected live, injured, and dead storm-petrels, and checked the bird shelters and surrounding areas during each survey. Live storm-petrels were assessed for injuries, measured, weighed, aged (darker colour and non-weathered condition of the primary feathers indicates juvenile birds [28]), and checked for the presence of a brood patch indicating breeding status [29]. On nights when 100–1,000 birds were stranded ("mass stranding events"), birds were only aged and counted. Most birds were released immediately from a dark area less than 1 km from the plant. Birds were released the following evening when they were injured, the weather was poor, or researcher availability was low. We recorded the total number of storm-petrels found, the number of live and dead birds, the lighting conditions at the plant (on or reduced), and cloud cover conditions (greater or less than 50%). The plant was closed for approximately two to three weeks during 2022 due to extraneous circumstances, meaning the lights were off and the lighting schedule was not followed. Therefore, the reduced light condition was disproportionately represented in this sample.

## Data analysis

**Daily morning and night surveys.**   Statistical analyses and figure construction were completed using R Statistical Software version 4.2.2 [30]. We assessed the period in which storm-petrels are most likely to strand by plotting the total number of stranded birds per day from the daily morning surveys (both alive and dead) in a time-series graph fit with a LOESS line of smoothing. We determined the peak stranding period as the time at which the smoothed line indicated that nightly strandings were double the mean number of stranded storm-petrels per night.

To determine whether reducing ALAN at the fish plant reduced the number of stranded storm-petrels, we used a negative binomial generalized linear model with a log link function. Using the daily morning survey data, we assessed the association of light condition (fixed; categorical; on or reduced), day of year, (fixed; continuous integer), year (fixed; categorical; 2021 or 2022), illuminated percent of the moon (hereafter moon illumination; fixed; continuous), wind speed (fixed; continuous), wind direction (fixed; categorical; onshore or offshore), and fog (fixed; categorical; fog or no fog) on the total number of storm-petrels stranded per night (live and dead). For 37 nights, data were unavailable for one of the variables, or morning surveys were completed following night surveys, influencing the results of the counts. As such, these observations were removed to proceed with model construction (data unavailable = 26, morning survey conducted following a night survey = 11, final sample size = 248 nights).

Model construction followed the guidelines recommended by Zuur et al. [31] and Zuur and Ieno [32] and were constructed using the package "glmmTMB" [33], with assumptions tested using the package "DHARMa" [34]. Moon illumination information was obtained with permission from the company Time and Date AS [35]. Wind speed (max gust per day) and direction information from a weather station in Grates Cove, NL (48.1675, -52.9393; 13 km from Bay de Verde, NL) were obtained from the Government of Canada website [36]. Wind direction was classified as onshore or offshore based on the tangent to the eastern Newfoundland coastline from Flowers Point on the Bonavista Bay Peninsula (48.59913, -52.99618) to Sugarloaf Head on the Avalon Peninsula (47.61971, -52.65213) (S3 Fig). Mass stranding events, in which hundreds to thousands of birds strand at one site in a single evening, are relatively uncommon but can have a disproportionate effect on the model. We therefore ran the analysis both with and without the outliers (large mass stranding events of $\geq$ 100 birds). The inclusion of outliers did not change the inference of the model, so analyses of the whole dataset are reported.

Night survey data were analysed using a similar negative binomial model as the one for the daily morning survey data, but adults and juvenile strandings were modelled separately and cloud cover (categorical, greater or less than 50%) was used in place of fog, as fog data were not available for this sample. Given that the earliest recording fledging date for the nearby colony of Great Island in Witless Bay, Newfoundland and Labrador, Canada was September 10 [29], we assumed any birds found before this date were adults. Since the majority of birds found during the night surveys after September 10 were juveniles, we assumed that any unaged birds found after this date were likely juveniles and included them in the juvenile model. The juvenile model also included date modelled as a quadratic term as juvenile strandings are not expected to have a linear relationship with day of year. Fledging occurs at a distinct period that follows a normal distribution with day of year and the number of chicks that fledge per night during this period is varied [37]. For three nights of the night surveys, data for one of the variables listed above were unavailable and had to be removed to proceed with model construction (final sample size for adult and juvenile models = 27 nights). If cloud cover data were not recorded, supplementary data from Custom Weather Inc [38] were used to estimate cloud cover conditions from the closest weather station in St. Johns, Newfoundland and Labrador.

**Ethical considerations.** Birds were collected under Canadian Wildlife Service scientific permit no. LS2688, and banded under Environment and Climate Change Canada Scientific Permits to Capture and Band Migratory Birds no. 10559 X and 10332 K.

## Results

### Effects of light and other environmental variables

### Daily morning surveys

The number of stranded birds per night was significantly ($p < 0.05$) greater when the lights were turned on, when moon illumination was low, and when winds were blowing onshore (Table 1). When the lights were reduced, 57.11% fewer birds tended to strand per night (95% CI: 39.29% - 83.01%). The predicted count from the model of stranded birds per night was over 1.5 times higher when the lights were on compared to when the lights were reduced (Fig 4). Significantly more birds stranded per night in 2021 than 2022 (Table 1; $M_{birdspernight2021}$ = 5.56, SD = 27.26; $M_{birdspernight2022}$ = 4.37, SD = 33.91). Day of year, wind speed, and fog were not significantly associated with the number of stranded birds. Variance in the number of Leach's Storm-Petrels stranded was significantly predicted by wind direction (Table 1).

**Table 1. Model coefficients, standard error values, and dispersion model of the negative binomial generalized linear model for the number of Leach's Storm-Petrels (*H. leucorhous*) found during daily morning surveys at a seafood processing plant in Bay de Verde, Newfoundland and Labrador, Canada in 2021 and 2022.**

|  | Estimate | Standard error | z-value | p-value |
|---|---|---|---|---|
| (Intercept) | 1.780 | 0.585 | 3.042 | 0.002 |
| Light condition (reduced) | -0.560 | 0.191 | -2.936 | 0.003 |
| Day of year | -0.004 | 0.002 | -1.892 | 0.058 |
| Year (2022) | -0.716 | 0.205 | -3.499 | < 0.001 |
| Moon illumination | -0.010 | 0.003 | -3.437 | < 0.001 |
| Fog (yes) | 0.154 | 0.199 | 0.776 | 0.438 |
| Wind speed | 0.008 | 0.006 | 1.317 | 0.188 |
| Wind direction (onshore) | 2.573 | 0.452 | 5.688 | < 0.001 |
| Dispersion model: | | | | |
|  | Estimate | Standard error | z value | p-value |
| (Intercept) | 1.559 | 0.222 | 7.016 | < 0.001 |
| Wind direction (onshore) | 3.550 | 0.538 | 6.597 | < 0.001 |

**Night surveys.** Similar to the daily morning surveys, the number of stranded adult birds per night during nocturnal surveys was significantly greater with the lights on and during 2021 (Table 2). When the lights were reduced, 11.94% fewer adult birds tended to strand per night (95% CI: 3.47% - 41.13%). The predicted count from the model of stranded adults per night was over 8 times higher when the lights were on compared to when the lights were reduced (Fig 5). Wind speed, wind direction, moon illumination, day of year, and cloud cover did not associate with the number of stranded adult storm-petrels per night. Variance in the number of stranded adult birds was not significantly predicted by the day of year (Table 2) but the inclusion of this term in the dispersion model improved model fit.

The number of stranded juvenile storm-petrels per night during night surveys was significantly greater when winds were blowing onshore and during 2021 (Table 3). As well, day of year had a quadratic relationship with the number of stranded juvenile birds (Table 3). Wind speed, cloud cover, and lighting condition did not associate with the number of stranded

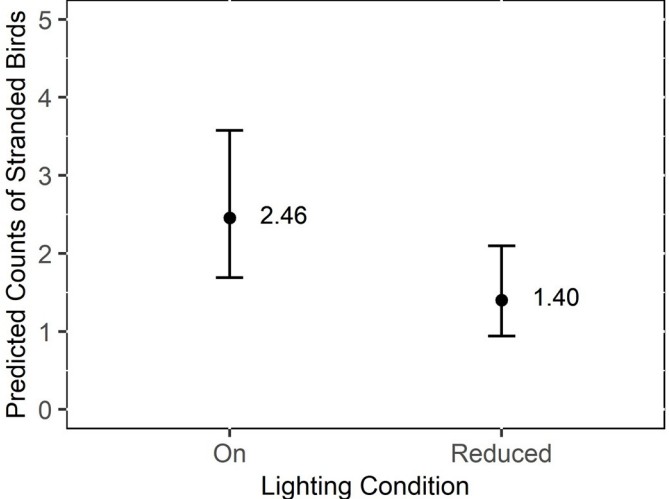

**Fig 4. Predicted count of Leach's Storm-Petrels (*H. leucorhous*) found during daily morning surveys at a seafood processing plant in Bay de Verde, Newfoundland and Labrador, Canada in 2021 and 2022 when the lights were on versus reduced.** The grey outline represents the 95% confidence interval.

**Table 2. Model coefficients, standard error values, and dispersion model of the negative binomial generalized linear model for the number of adult Leach's Storm-Petrels (*H. leucorhous*) found during night surveys at a seafood processing plant in Bay de Verde, Newfoundland and Labrador, Canada in 2021 and 2022.**

|  | Estimate | Standard error | z-value | p-value |
|---|---|---|---|---|
| (Intercept) | 0.269 | 1.652 | 0.163 | 0.871 |
| Light condition (reduced) | -2.125 | 0.631 | -3.368 | < 0.001 |
| Day of year | 0.008 | 0.005 | 1.469 | 0.142 |
| Year (2022) | 2.508 | 1.164 | 2.154 | 0.031 |
| Moon illumination | -0.004 | 0.006 | -0.690 | 0.490 |
| Cloud cover (> 50%) | -0.306 | 0.790 | -0.387 | 0.699 |
| Wind speed | -0.014 | 0.026 | -0.540 | 0.589 |
| Wind direction (onshore) | -0.160 | 0.452 | -0.353 | 0.724 |
| Dispersion model: |  |  |  |  |
|  | Estimate | Standard error | z-value | p-value |
| (Intercept) | 3.327 | 2.354 | 1.414 | 0.157 |
| Day of year | -0.018 | 0.048 | -0.376 | 0.707 |

juveniles. Though moon illumination was not significant, there was a trend indicating the number of stranded juvenile birds per night increased with less moon illumination (Table 3).

**Peak stranding period.** Consistent with findings from the night survey analysis that indicated a quadratic relationship between day of year and the number of juvenile birds stranded per night, the peak stranding period occurred from September 25 to October 28 (Fig 6), when the LOESS line of smoothing indicated that the nightly strandings were consistently greater than double the mean number of birds that stranded per night (M = 4.25, SD = 27.67). This interval encompassed 83% of all birds stranded.

## Mortality, survival, and age class

In total, including observations that were removed for model construction, 3,925 Leach's Storm-Petrels were found stranded at the seafood processing plant in 2021 and 2022. Of these

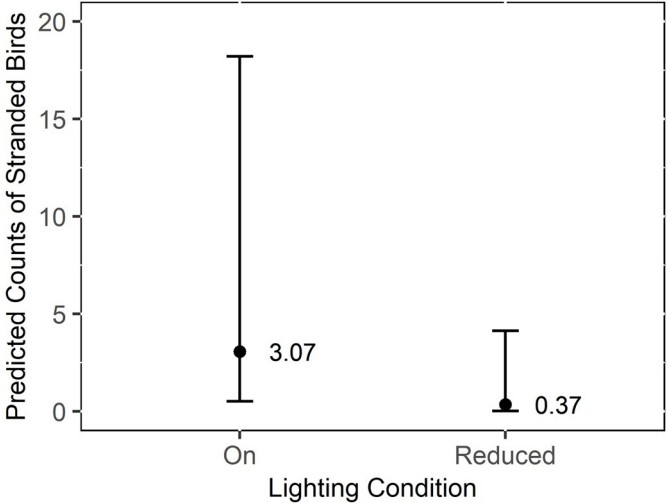

**Fig 5. Predicted count of adult Leach's Storm-Petrels (*H. leucorhous*) found during night surveys at a seafood processing plant in Bay de Verde, Newfoundland and Labrador, Canada in 2021 and 2022 when the lights were on versus reduced.** The grey outline represents the 95% confidence interval.

**Table 3. Model coefficients and standard error values of the negative binomial generalized linear model for the number of juvenile Leach's Storm-Petrels (*H. leucor-hous*) found during night surveys at a seafood processing plant in Bay de Verde, Newfoundland and Labrador, Canada in 2021 and 2022.**

|  | Estimate | Standard error | z-value | p-value |
|---|---|---|---|---|
| (Intercept) | 17.233 | 9.550 | 1.804 | 0.071 |
| Light condition (reduced) | -0.486 | 0.759 | -0.640 | 0.522 |
| Day of year | -0.190 | 0.085 | -2.240 | 0.025 |
| Day of year (quadratic) | 0.001 | 0.000 | 2.843 | 0.004 |
| Year (2022) | -2.795 | 1.127 | -2.479 | 0.013 |
| Moon illumination | -0.013 | 0.008 | -1.650 | 0.099 |
| Cloud cover (> 50%) | -0.118 | 0.625 | -0.188 | 0.851 |
| Wind speed | 0.005 | 0.026 | 0.188 | 0.851 |
| Wind direction (onshore) | 1.871 | 0.664 | 2.818 | 0.005 |

birds, 92.7% were alive. In 2021 and 2022, across all surveys, 1,950 birds stranded when the building LEDs were on compared to 1,929 birds when the lights were reduced. In total, across both years, the number of birds found during the night surveys (n = 2,590 birds, mean = 74 birds per night; Table 4) was nearly double the number of birds found during the daily morning surveys (n = 1,335 birds, mean = 4.3 birds per night; S2 Fig). Most adults stranded from June to August (65.6%), and juveniles composed most (94.9%) of the birds found during the peak stranding period (Table 5). Of the birds we were able to collect and assign age to in 2022 (live and dead), 361 were juveniles and 212 were adults.

**Mass stranding events.** Eight mass stranding events (≥ 100 birds stranding during a single night) occurred during the 2-year data collection period and all occurred within the peak stranding period; September 30 2021, October 1–4 2021, October 16, 2022, and October 19–20, 2022. Of these mass stranding events, five occurred when the lights were reduced. Notably, during six of these events, moon illumination only reached a high of 38%, cloud coverage was

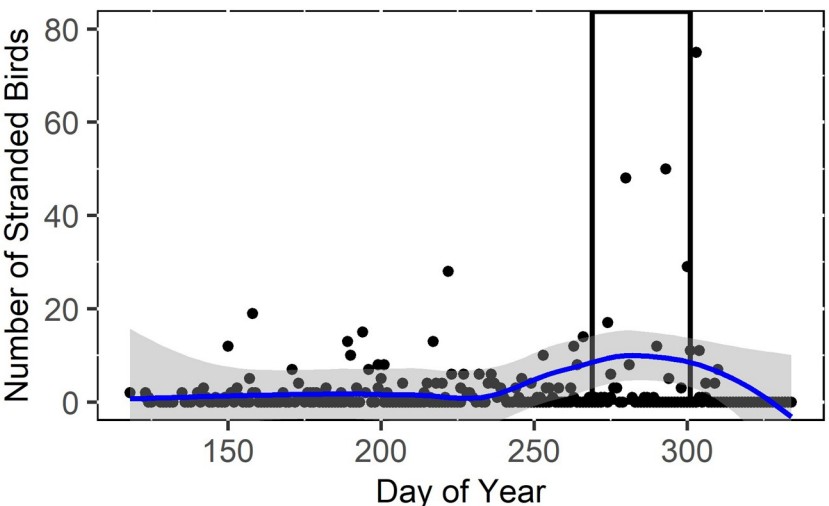

**Fig 6. Scatterplot of the number of stranded Leach's Storm-Petrels (*H. leucorhous*) per day of year in 2021 and 2022 collected during the daily morning surveys of a seafood processing plant in Bay de Verde, Newfoundland and Labrador, Canada (day 250 = 7 September).** The blue line is the loess line of smoothing, the grey surrounding the line is the 95% confidence interval, and the black rectangle represents the peak stranding period (September 25 - October 28). Two outliers (number of stranded birds = 387 and 283) were used to create the graph but are not shown on the above figure to improve data visualization.

**Table 4. Summary of age, breeding status and individual measurements of stranded Leach's Storm-Petrels (*H. leucorhous*) collected during night surveys at the seafood processing plant in Bay de Verde, Newfoundland and Labrador, Canada, during 2021 and 2022.**

| Parameter | Value |
|---|---|
| Number of adults | 250 |
| Number of juveniles | 1,609 |
| Number of unknown-age birds[a] | 731 |
| Number of breeding individuals[b] | 59 |
| Mean adult weight and standard deviation (n = 133)[b] | 41.82 g +/- 11.11 g |
| Mean juvenile weight and standard deviation (n = 13)[b] | 45.54 g +/- 6.89 g |
| Mean adult wing length and standard deviation (n = 135)[b] | 160.21 mm +/- 3.58 mm |
| Mean juvenile wing length and standard deviation (n = 15)[b] | 159.00 mm +/- 3.27 mm |

[a]Birds of unknown age were either unable to be identified by the handler, escaped capture, or were not aged or measured due to logistical constraints.

[b]Measurements of breeding status, wing length, and weight were only available for 2022.

greater than 50% or it was foggy, and winds did not fall below 35 km/hr. Six mass stranding events occurred during the night surveys. During these events, juveniles comprised 66% of the birds, adults comprised only 4%, and 30% were unknown-age.

## Discussion

We experimentally demonstrated that even a partial reduction in coastal lighting was an effective mitigation strategy for birds stranding around artificially lit areas, reducing strandings by 11 to 57% (Tables 1 and 2). Our results concur with those of Miles et al. [6] who conducted a light experiment with Leach's Storm-Petrels, and reported a decrease in the number of strandings when the street lights at a common stranding area were turned off, compared to the previous year with which the lights were turned on. We also demonstrated that varying the lighting condition both within and across years yields similar findings, suggesting that Leach's Storm-Petrels exhibit positive phototaxis. It is well established that other burrow-nesting procellariiforms cluster around ALAN [5, 22, 23, 39], and in some species, reducing ALAN by turning off or shielding lights has been effective [22, 23]. For example, Reed et al. [23] showed that shielding lights at a common stranding hotspot on Kauai Island, Hawaii decreased the number of grounded Newell's Shearwaters (*Puffinus newelli*) recovered by almost 40%. On Phillip Island, Australia where more than 8,000 Short-tailed Shearwaters (*Ardenna tenuirostris*) were grounded from 1999 to 2013, turning off road lights on a brightly illuminated bridge decreased the number of stranded shearwaters [22].

In addition to ALAN, moon illumination was a key environmental factor which influenced strandings in the current study. Morning surveys revealed that fewer storm-petrels stranded when moon illumination was high (Table 1). During night surveys, moon illumination was not significantly associated with storm-petrel strandings in either adult nor juvenile models, however there was a trend in juvenile birds towards a reduction in strandings during periods of high moon illumination (Table 3). Numerous studies show that greater moon illumination is associated with decreased strandings and reduced colony activity by adult Leach's Storm-Petrels and other procellariiforms [4, 5, 10, 21, 23]https://www.zotero.org/google-docs/?1beLPI. This phenomenon has two proposed explanations: 1) lunar illumination appears to reduce the disorienting influence of ALAN [21], and 2) birds are less active on nights with high moon illumination [40], so fewer strandings occur. Though adults tend to be less active

**Table 5. Total number of adult, juvenile, and unknown-age Leach's Storm-Petrels (*H. leucorhous*) that stranded at the seafood processing plant in Bay de Verde, Newfoundland and Labrador, Canada, per night during each month of night survey data collection in 2021 and 2022.**

| Year | Age | Month | | | | | | |
|------|-----|-------|-----|------|------|--------|-----------|---------|
| | | April | May | June | July | August | September | October |
| 2021 | Adult | _a | _a | 8 | _a | _a | 16 | 14 |
| | Juvenile | _a | _a | 0 | _a | _a | 24 | 1,224 |
| | Unknown-age | _a | _a | 0 | _a | _a | 0 | 684 |
| 2022 | Adult | 0 | 0 | 46 | 41 | 70 | 9 | 46 |
| | Juvenile | 0 | 0 | 0 | 0 | 0 | 32 | 329 |
| | Unknown-age | 0 | 0 | 0 | 0 | 0 | 8 | 39 |

[a]Surveys were not completed during these months.

on a full moon [40], Collins et al. [37] found that the fledging of juvenile Leach's Storm-Petrels is not associated with moon phase or incident illumination, suggesting that fledging activity does not adequately predict strandings when moon illumination is high [25, 37]. Given that fewer birds in our study stranded on a full moon and that most collected in our study were fledglings (Table 4), it is likely that moonlight reduces the risk for stranding by reducing the attractive and disorientating properties of ALAN [25, 37].

Wind direction was also associated with the number of stranded juvenile birds per night, suggesting that because juveniles are less experienced flyers, they are susceptible to being pushed toward land by onshore winds [10, 29]. Contradicting previous research [11], fog, cloud cover, and wind speed did not associate with the number of stranded birds, however, five out of eight mass stranding events (63%) during our study occurred when moon illumination was low, cloud cover was high or it was foggy, and winds were blowing onshore. These results suggest that fog and cloud cover, environmental variables which influence available nocturnal illumination, may play a role in mass stranding events. To make better-informed assessments of the effect of cloud cover and fog, more research is needed into these episodic occurrences.

In 2021 and 2022, researchers and volunteers rescued more than 3,500 stranded Leach's Storm-Petrels at the seafood processing plant in Bay de Verde. Despite daily monitoring, hundreds of birds perished (see body part count analysis S1–S3 Texts). The number of birds found during night surveys was almost double the number found when conducting morning surveys, even though night surveys only accounted for 10% of the total surveying effort (S1 Fig). Daily morning surveys, while often the only feasible option, do not account for birds that were removed by predators or died in inaccessible locations. To maximize the survival of stranded Leach's Storm-Petrels, consistent researcher or volunteer presence is needed at common stranding areas during the peak stranding period to conduct nightly surveys.

The peak stranding period of birds at Bay de Verde occurred from September 25 to October 28 (Fig 6) and night surveys showed that the number of stranded juvenile birds had a quadratic relationship with day of year (Table 3). These results concur with previous research which found that most storm-petrels stranded during late September through to November in Atlantic Canada [3, 4, 10, 12, 13]. This period corresponds directly with the Leach's Storm-Petrels' fledging period [29, 37] and most birds found during this period were fledglings (Table 5), supporting the idea that they are the age class most susceptible to the effects of ALAN [5, 9, 41]. The eight mass stranding events that occurred during this study consisted of mostly fledglings, but did not coincide with the lights on condition, as five out of eight (63%) of the mass stranding events occurred when the lights at the plant were reduced. This proportion is similar

to the proportion of nights in our study when the lights were reduced (61%). One explanation for the occurrence of mass stranding events relative to light condition is that fledglings exhibit stronger positive phototaxis and more intense disorientation relative to adults [10, 13], and a partial reduction in lighting is not enough to eliminate strandings.

While the loss of fledglings is concerning, the hundreds of adults, 59 of which were verified breeders based on brood patch presence (Table 4), raises concerns for population-level effects. Most verified breeders stranded from June to August. The number of stranded verified breeders may have been underestimated, as brood patch feathers begin to re-grow four weeks after hatching (approximately 20 to 30 July for the largest nearby colony on Gull Island in Witless Bay, NL [27]), making it difficult to determine breeding status after this period. The mortality of breeding adults is concerning because storm-petrels are serially monogamous, exhibit delayed maturity, breed once a year, and lay a single egg annually [29]. When a breeding adult dies, it disrupts a long-term monogamous pair bond [27] and their egg or chick will likely die [42], as seen in other long-lived monogamous seabird species [43]. Though individuals can form new pair bonds, new pairs may take time to re-establish and often have poor reproductive success for several years [44]. Even though most birds strand from September to October, it is important to reduce nocturnal lighting throughout the breeding season to protect breeding adults as well as juveniles. While the mortality associated with ALAN is not known to impact Leach's Storm-Petrel populations, the species' precarious state warrants efforts to curtail all aspects of human-induced mortality.

Knowledge of the timeline and locations of when and where storm-petrels are most likely to strand allows for a fine-tuning of conservation efforts related to positive phototaxis. A recent study examined social media reports of Leach's Storm-Petrel strandings across the island of Newfoundland and found that the majority of birds stranded in brilliantly illuminated urban areas [13], suggesting that ALAN is likely causing widespread strandings throughout insular Newfoundland and the species' breeding range (see also [10]). Conservation efforts at the seafood processing plant, including turning off the lights, using bird shelters, and searching daily for stranded birds to reduce mortality have been employed elsewhere and are effective (e.g., Newfoundland and Labrador Hydro Thermal Generation Station), though these efforts are not always feasible. For example, vessels arrive throughout the night at the plant in Bay de Verde and the lights must be on to ensure safe visibility when unloading catch. Furthermore, bird shelters must be checked daily to retrieve live birds. Therefore, it is important to seek out other permanent solutions to reduce strandings by exploring the properties of light that influence phototaxis such as peak wavelength, light source, intensity, and the size of the light catch basin [23, 45, 46]. Contention surrounds the attractive properties of different wavelengths of light. Some research suggests that blue and green light attract the most migratory passerines in cloudy conditions, while in clear conditions there was no effect for wavelength [47]. Other results showed that all other wavelengths, except red, deterred adult Manx Shearwaters (*Puffinus puffinus* [46]). The type of light source can also have an impact; high-pressure sodium lights attract fewer shearwaters than metal halide lights and LEDs [45]. Specific research investigating the wavelengths and light types to which Leach's Storm-Petrels are the most attracted has not been conducted. Future research should focus on determining the spectral sensitivities of adult Leach's Storm-Petrels (but see Mitkus et al. [41]) to inform decisions about mitigations to reduce population mortality. Shielding light has been successful for limiting nocturnal positive phototaxis by other procellariiforms, because shielding redirects the light to reduce the light projected skyward [23]. Similarly, motion-activated lights should be considered in areas where constant lighting is unnecessary [46].

Increasing rescue efforts at hot-spot stranding locations where near-zero lighting cannot be achieved is also recommended to maximize recovery of stranded storm-petrels. Performing

systematic nightly searches during the peak stranding period (from late September to the end of October) and deploying an array of bird shelters at the start of the breeding season (early May) that are checked daily can reduce mortality of breeding adults and recently fledged juveniles at common stranding locations. This strategy is currently used by the "Puffin and Petrel Patrol" program [48], executed by a non-profit organization, the Canadian Parks and Wilderness Society–Newfoundland and Labrador, that provides volunteers with training on how to rescue stranded fledgling Atlantic Puffins (*Fratercula arctica*) and storm-petrels, and subsequently sends volunteers to common stranding locations in southeastern Newfoundland during the night. Storm-petrels strand across a large geographical scale [4, 10, 13] and widespread public education regarding light-induced storm-petrel and other seabird strandings coupled with the development of volunteer rescue groups throughout their breeding range could dramatically reduce seabird mortality associated with attraction to ALAN.

## Conclusion

The evidence indicates that turning off extraneous lights can significantly decrease Leach's Storm-Petrel strandings and that even partial light reductions are effective in this regard. More than 4,000 storm-petrels were found stranded at a seafood processing plant in Newfoundland and Labrador in 2021 and 2022. This stranding estimate is likely highly conservative, as it does not include mortality that occurred throughout the beginning of the breeding season when researchers were not present, nor does it include mortality that occurred throughout the night as a result of predators removing evidence of predation on stranded birds. Increasing research and rescue efforts and promoting public education about Leach's Storm-Petrels and all seabirds that strand near ALAN is important, but employing a preventative strategy to mitigate seabird strandings is a more efficient and effective solution. Therefore, we recommend shifting conservation efforts to focus on reducing unnecessary extraneous sources of ALAN as much as possible, particularly during the peak stranding period.

## Supporting information

**S1 Fig. The number of daily morning surveys (grey) and night surveys (black) performed each month in 2021 and 2022 at the seafood processing plant in Bay de Verde, Newfoundland and Labrador, Canada.**
(TIF)

**S2 Fig. The number of Leach's Storm-Petrels stranded per night during 2021 and 2022 found during the daily morning surveys (square points) and night surveys (triangle points) when the lights at the seafood processing plant in Bay de Verde, Newfoundland and Labrador, Canada were on (dark blue) or reduced (light blue).**
(TIF)

**S3 Fig. A section of the eastern Newfoundland coastline with red points indicating Flowers Point, Bay de Verde, and Sugarloaf Head.** The bearing angle between Flowers Point (166˚) and Sugarloaf Head (346˚) was calculated relative to North (0˚) using the black tangent line. This angle was used to classify wind direction as either onshore (less than 166˚ and greater than 346˚) or offshore (greater than 166˚ and less than 346˚).
(TIF)

**S1 Text. Body part count data collection.**
(DOCX)

**S2 Text. Body part count data analysis.**
(DOCX)

**S3 Text. Body part count results.**
(DOCX)

## Acknowledgments

We thank Robin Quinlan, Barry Hatch, Kristinn Skulason, Ed and Cindy Noonan, Jim and Cheryl Broderick, and all of the staff at the Quinlan Brothers Ltd. seafood processing plant for permitting and supporting the project, and for assisting with the light schedule and bird collection. Thank you to Robert Blackmore, Taylor Brown, Juliana Coffey, Kyle D'Entremont, Mohammad Fahmy, Gretchen McPhail, Fiona Le Taro, and Christopher Ward for help collecting and releasing birds. Thank you to Alex Day for assistance with data analysis. We thank Time and Date AS and Custom Weather Inc. for providing weather data. We are grateful to Jake Russell-Mercier of ECCC for continued diligence and efforts to help fund this research. We thank reviewers Airam Rodríguez and Taylor Brown, and editor Travis Longcore for their comments and considerations which greatly improved the manuscript.

## Author Contributions

**Conceptualization:** Tori V. Burt, Sydney M. Collins, Sherry Green, Sabina I. Wilhelm, William A. Montevecchi.

**Data curation:** Tori V. Burt, Sydney M. Collins.

**Formal analysis:** Tori V. Burt, Sydney M. Collins.

**Funding acquisition:** Sabina I. Wilhelm, William A. Montevecchi.

**Investigation:** Tori V. Burt, Sherry Green, Parker B. Doiron, Sabina I. Wilhelm, William A. Montevecchi.

**Methodology:** Tori V. Burt, Sydney M. Collins, Sherry Green, Parker B. Doiron, Sabina I. Wilhelm, William A. Montevecchi.

**Project administration:** Tori V. Burt, William A. Montevecchi.

**Resources:** Sabina I. Wilhelm, William A. Montevecchi.

**Supervision:** William A. Montevecchi.

**Validation:** Tori V. Burt, Sydney M. Collins, Parker B. Doiron.

**Visualization:** Tori V. Burt, Sydney M. Collins, Parker B. Doiron.

**Writing – original draft:** Tori V. Burt, Sydney M. Collins, William A. Montevecchi.

**Writing – review & editing:** Tori V. Burt, Sydney M. Collins, Parker B. Doiron, Sabina I. Wilhelm, William A. Montevecchi.

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
