## [Decision Letter · Decision Letter 0]

27 Dec 2023

PONE-D-23-37663Reduction of coastal lighting decreases seabird strandingsPLOS ONE

Dear Dr. Burt,

Thank you for submitting your manuscript to PLOS ONE. After careful consideration, we feel that it has merit but does not fully meet PLOS ONE’s publication criteria as it currently stands. Therefore, we invite you to submit a revised version of the manuscript that addresses the points raised during the review process. Please consider carefully the detailed comments from both reviewers.  It is critical that the data depository work be functional.  Please respond specifically to the questions regarding the statistical approach by reviewer 1 (whether to model juveniles separately, using a quadratic term for date).  I believe that you will find the comments to all be useful, productive, and thorough.  The second reviewer already disclosed a potential conflict resulting from collaboration with one author, so there is no need for you to worry about that (the review is signed).

We look forward to receiving your revised manuscript.

Kind regards,

Travis Longcore, Ph.D.

Academic Editor

PLOS ONE

2. We noted in your submission details that a portion of your manuscript may have been presented or published elsewhere. [The daily morning survey data were used to create Figure 2 in a pending publication manuscript in Biology Letters titled "Small tube-nosed seabirds fledge on the full moon and throughout the lunar cycle." These data were not used in any analyses and were only included in Figure 2 to show stranding occurrences relative to moon illumination and fledgling in Leach's Storm-Petrels.] Please clarify whether this publication was peer-reviewed and formally published. If this work was previously peer-reviewed and published, in the cover letter please provide the reason that this work does not constitute dual publication and should be included in the current manuscript.

4. In the online submission form, you indicated that [Insert text from online submission form here]. 

5. We note that Figure 3 in your submission contain copyrighted images. All PLOS content is published under the Creative Commons Attribution License (CC BY 4.0), which means that the manuscript, images, and Supporting Information files will be freely available online, and any third party is permitted to access, download, copy, distribute, and use these materials in any way, even commercially, with proper attribution. For more information, see our copyright guidelines: http://journals.plos.org/plosone/s/licenses-and-copyright.

a. You may seek permission from the original copyright holder of Figure 3 to publish the content specifically under the CC BY 4.0 license. 

6. We note that Figure(s) 1 and S2 in your submission contain [map/satellite] images which may be copyrighted. All PLOS content is published under the Creative Commons Attribution License (CC BY 4.0), which means that the manuscript, images, and Supporting Information files will be freely available online, and any third party is permitted to access, download, copy, distribute, and use these materials in any way, even commercially, with proper attribution. For these reasons, we cannot publish previously copyrighted maps or satellite images created using proprietary data, such as Google software (Google Maps, Street View, and Earth). For more information, see our copyright guidelines: http://journals.plos.org/plosone/s/licenses-and-copyright.

a. You may seek permission from the original copyright holder of Figure(s) 1 and S2 to publish the content specifically under the CC BY 4.0 license.  

Reviewers' comments:

Reviewer's Responses to Questions

**Comments to the Author**

1. Is the manuscript technically sound, and do the data support the conclusions?

Reviewer #1: Yes

Reviewer #2: Yes

2. Has the statistical analysis been performed appropriately and rigorously? 

Reviewer #1: I Don't Know

Reviewer #2: Yes

3. Have the authors made all data underlying the findings in their manuscript fully available?

Reviewer #1: No

Reviewer #2: No

4. Is the manuscript presented in an intelligible fashion and written in standard English?

Reviewer #1: Yes

Reviewer #2: Yes

5. Review Comments to the Author

Reviewer #1: In this paper, the authors assess the impact of lighting management (turning the main lights off) in the fallout numbers of Leach’s storm petrels at a seafood processing plant in Canada. I feel the ms is worthy of publication, but I identify some weak points regarding to the data analyses that might be addressed by the authors (see below).

Major comments

Data analyses

1) You include the number of adults and fledglings as a response variable. However, response to light is different between adults and fledglings, but also depending on the type of bird activity (foraging, visiting the colony at flight, fledging or when they are on land). See different responses for adults, nestlings and fledglings in, for example, Syposz et al. 2021 Scientific Reports, Rodríguez et al 2022 Frontiers in Ecology and Evolution or Atchoi et al. 2023 Journal of Experimental Biology. So, my suggestion is to build separate models for adults and fledglings. Given that you might have some unaged birds during the fledging period, you could run two models: one conservative using just the fledglings, and another including fledglings + unaged birds. I bet the influence of explanatory variables will be the same. Alternatively, you might assume that all grounded birds during the fledging period are fledglings (95%!! According to your data in line 285).

2) You use GLMMs and include day of the year as a predictor, but you are not expecting a linear relationship (showed in figure 6). If you follow my previous suggestion, in the case of fledglings, you should use a quadratic term as done in Rodríguez et al 2014 Plos One. It is reasonable to think that fledging follows a quadratic term along the date. In the case of adults, you might include the simple term ‘day of the year’ as long as you are expecting that the number of grounded adults decline with the date.

3) What is the random term in your GLMM? It is not stated in lines 193-211. Then in results, you have some figures from negative binomial generalized models, but they have not been explained in Methods. Is it a misunderstanding between GLMMs and GLMs?

4) Mortality analyses. I am not sure why authors do not model mortality as a proportional response variable with a binomial model. You have the numbers of dead and alive birds, so authors can identify the variables affecting mortality rates. I would also separate adults from fledglings as variables influencing mortality rates might be different for both groups. See Rodríguez et al 2014 Plos One for a similar statistical procedure.

Widening the scope of the ms

Introduction is focussed on Leach’s storm petrel, which is great, but why do you think that Leach’s storm-petrel is going to be so different to other seabird species? Given that seabird attraction to light is a common phenomenon to underground nesting seabirds (and with multiple similarities among the species involved), I would try to widen the scope to seabirds in general (you already did it in the title!). You might include all the specific information of the Leach’s storm petrel (e.g. lines 51-58 or 61-62), in a section entitled Model species or Study species (or something like that). Then, in lines 59-83, every time you talk about storm-petrels you might interchange by seabirds (at least, in most cases).

Minor comments

Lines 50-51: You cite here Reed et al 1985 and Wilhelm et al 2021, but they don’t provide a list of species affected. In this sense, I feel more appropriate to state in this sentence that Procellariiformes is the most vulnerable with the number of affected species. In this sense, the most updated reference is the Chapter by Gilmour et al 2023 in the book Conservation of Marine Birds.

Lines 71-72: I disagree with this sentence. There are at least another two studies where light reduction has reduced fallout and you have them included in the reference list: One is the pioneer study by Reed et al 1985 running a very similar experiment to yours, another is Rodríguez et al 2014 where fallout was reduced by turning the lights of a bridge off. I think that this information should be showed to readers in the introduction (as authors have very well done with the study by Miles et al. 2010).

Lines 229-230: According to Table 1, fog is significant! Day of year may not be significant because the reasons given above.

Table 1. I do not understand why there are no estimates for all predictors nor why the dispersion model only includes fog. Are these the results of the GLMM?

Line 280. It is the first time you call LEDs. It would be great to have a better description of the type of lights were on/off during the experiment. Critical aspects are intensity of the lamps and spectrum.

Table 3. How do you assess the breeding status to complete the row ‘N of breeding individuals’? Should that information be at Methods?

Lines 306-321: The think that the first paragraph is out of the scope of the ms: 1) I miss a discussion of the Reed et al 1985 study! 2) Lines 314-317, Telfer and co-workers study is not an experiment! 3) Lines 319: “…suggesting that they also exhibit positive phototaxis.” It seems like authors are re-inventing the wheel. I think that there is a great consensus on that seabird fledglings show phototaxis in flight. This has been demonstrated by GPS-tracking flights in Rodríguez et al 2015 Scientific Reports and 2022 Frontiers in Ecology and Evolution. In summary, I think that this first paragraph should be focussed on the reductions of fallout numbers by turning the lights off or shielding the lights described by Miles et al., Reed et al. and Rodríguez et al. (not sure if I am missing other critical studies).

Lines 327-330: Although it might be useful for adults, please, note that the explanation about the reduced activity of fledglings (fledging) because high moon illumination is not a valid hypothesis, at least until new evidence arise. A full explanation is available in Rodríguez et al 2023 Conservation Science and Practice. So, please, reword to address the differences between age groups.

Lines 336-339: First, note that fog is significant according to table 1. Second, it seems that the second part of the sentence is contradictory to the first one.

Lines 339-340: I would rewrite this part “ …most mass stranding events (63%)…” to something like “…six out of eight mass stranding events…” Confirm the numbers. Also about these lines and followings, could you give some information on the weather conditions? For example, was it rainy, cloudy or foggy during those six nights of mass stranding events?

Lines 359: “…suggesting that…” What is the doubt? Since long time ago, it has been widely accepted that fledglings are more affected than adults.

Lines 361: Be consistent with previous lines. 63% or 62.5%

Lines 362-365: Sorry, I don’t follow the thread or your reasoning. Why are you proposing these two hypotheses? What is the evidence to propose the second hypothesis?

Lines 372: Great!! “…based on brood patch presence.” This should be stated in Methods.

Lines 379: I would say that no eggs or nestlings would survive if one parent die. There should be plenty of references on petrels.

Lines 399-401: “…while other results suggest that red and white light attract the most seabirds and migratory birds and cause the most disorientation in foggy conditions (43,46).” Please caution here! First, Syposz et al 2021 show that ADULT manx shearwaters are more deterred with white and blue lights (not attracted!). Adult is in capital letters to highlight the differences between adults and fledglings, which support my proposal of two models to each age group. Second, Poot et al. 2008 has been replied by Evans et al 2010 (https://ecologyandsociety.org/vol15/iss3/resp1) because of their experimental design and Poot and co-workers never replied. In addition, most evidence published after that paper indicate the opposite. So, I think that paper should not be cited anymore.

Lines 408-409: Consider citing here Syposz et al 2021 as they found a smaller effect with shorter light treatments. To my best knowledge, it is the only paper testing that.

Lines 413-415: Deploying an array of bird shelters deserve a paper explaining their potential use, pros and cons!! If you are not going to do it, I would ask for providing some information in this contribution on what is the proportion of birds found in the shelters vs the birds collected out, alive vs dead, etc. It seems as a potentially useful mitigation of light-induced mortality.

In the next Review Questions, I would like to explain my selected responses:

2. Has the statistical analysis been performed appropriately and rigorously?

I don’t know because authors are using GLMs or GLMMs. Futhermore, I feel that they should run different models for adults and fledglings (see above).

3. Have the authors made all data underlying the findings in their manuscript fully available?

I have selected No because the GitHub link does not work.

I hope authors find useful these comments to improve its paper.

Reviewer #2: Please see attached.

6. PLOS authors have the option to publish the peer review history of their article (what does this mean?). If published, this will include your full peer review and any attached files.

Reviewer #1: **Yes: **Airam Rodríguez

Reviewer #2: No

---

## [Author Response · Author response to Decision Letter 0]

17 Feb 2024

We thank the reviewers for their constructive comments and for their time and efforts in reviewing the manuscript. Our responses to their comments have been provided in a document titled "Response to Reviewers" attached to the submission.

---

## [Decision Letter · Decision Letter 1]

24 Mar 2024

PONE-D-23-37663R1Reduction of coastal lighting decreases seabird strandingsPLOS ONE

Dear Dr. Burt,

Thank you for submitting your manuscript to PLOS ONE. After careful consideration, we feel that it has merit but does not fully meet PLOS ONE’s publication criteria as it currently stands. Therefore, we invite you to submit a revised version of the manuscript that addresses the points raised during the review process. One of the reviewers has some constructive suggestions for a very minor revision that I ask that you consider.

We look forward to receiving your revised manuscript.

Kind regards,

Travis Longcore, Ph.D.

Academic Editor

PLOS ONE

Journal Requirements:

**Additional Editor Comments:**

Please consider a very minor revision to address some remaining comments and suggestions.

Reviewers' comments:

Reviewer's Responses to Questions

**Comments to the Author**

1. If the authors have adequately addressed your comments raised in a previous round of review and you feel that this manuscript is now acceptable for publication, you may indicate that here to bypass the “Comments to the Author” section, enter your conflict of interest statement in the “Confidential to Editor” section, and submit your "Accept" recommendation.

Reviewer #2: (No Response)

2. Is the manuscript technically sound, and do the data support the conclusions?

Reviewer #2: Yes

3. Has the statistical analysis been performed appropriately and rigorously? 

Reviewer #2: Yes

4. Have the authors made all data underlying the findings in their manuscript fully available?

Reviewer #2: Yes

5. Is the manuscript presented in an intelligible fashion and written in standard English?

Reviewer #2: Yes

6. Review Comments to the Author

Reviewer #2: Review of Burt et al.: Reduction of coastal lighting decreases seabird strandings

Taylor Brown

I appreciate the opportunity to re-evaluate this manuscript; it was a pleasure to read. The authors have done an excellent job of responding to my queries, and I find this draft to be vastly improved from the previous one, thanks also in large part to the suggestions of the other reviewer. The new, separate models for juveniles and adults help to further clarify trends in the drivers of stranding for each age group. I have no major comments, and include only a handful of minor suggestions below.

Please note that line numbers refer to the clean edited version of the manuscript.

L108-13. Here and in Figure 2, it still appears that some of the lights along the northwest side (the “front”) of the building that were un-manipulated are not mentioned or included in the overall count of how many lights are present in the immediate area of the building. Specifically, in the “reduced” lighting photo (Figure 3b) I can see five pairs of LED lights (so ten total) along the wharf, plus a cluster of four bright LED lights on the far right side of the building and two apparent High Pressure Sodium pole lights in the middle of the picture beside the wharf.

L241-42. Fog was also not significantly associated with the number of stranded birds and could be added to this sentence.

L254-62. Thank you for specifying adults in L254. It may help the reader if the rest of the paragraph also refers specifically to “stranded adults” or “stranded adult birds” rather than the more general “stranded birds” or “stranded storm-petrels”. Similarly, in L272 juveniles are specified but the rest of the paragraph could also refer specifically to juveniles for utmost clarity (L272-77).

Table 4. Please clarify in the table caption if means are accompanied by standard deviation.

L346-48. I suggest clarifying that “fewer storm-petrels of [all age classes] stranded when moon illumination was high…” and either remove or change the wording of the second clause which refers to juveniles specifically, because of the non-significance of that result. The morning survey data (indicating a significant effect of moon illumination) have much higher resolution than the night survey data anyway, so greater emphasis / confidence could be placed on that result.

L352-55. The grammatical structure of this sentence could be improved slightly for readability.

L357. Perhaps start a new paragraph here for wind and other weather effects.

L386-90. Is it possible to somehow combine this section with L365-69, or condense it, since both sections cover the potential causes of mass strandings and how they relate to light condition? This could be achieved, in part, by shifting the intervening paragraph (L370-78) down so that it immediately precedes the paragraph that currently starts at L391 (about the importance of adult mortality). This flow of ideas would seem to me to be more logical. Also, L388-89 appears to imply that a potential explanation for the random nature of mass stranding events (with respect to lighting condition) is due to a stronger phototactic response by fledglings; but if it is random, then I would assume there is no light-related explanation.

L442. This is the first time that “puffin” appears in the manuscript, so I suggest including the binomial species name.

Figure S2A. There appear to be two triangular (night survey) points in October that lack colour to indicate lighting condition. Is this a mistake, or is light condition unknown? Overall, I very much appreciate the addition of this figure to the supporting information! It’s nicely done.

7. PLOS authors have the option to publish the peer review history of their article (what does this mean?). If published, this will include your full peer review and any attached files.

Reviewer #2: **Yes: **Taylor Brown

---

## [Author Response · Author response to Decision Letter 1]

24 Apr 2024

Please see the attached "response to reviewers" document for comments.

---

## [Editor Report · Decision Letter 2]

2 May 2024

Reduction of coastal lighting decreases seabird strandings

PONE-D-23-37663R2

Dear Dr. Burt,

We’re pleased to inform you that your manuscript has been judged scientifically suitable for publication and will be formally accepted for publication once it meets all outstanding technical requirements.

Kind regards,

Travis Longcore, Ph.D.

Academic Editor

PLOS ONE

Additional Editor Comments (optional):

Thank you for your patience and diligence with revisions.
---

## [Editor Report · Acceptance letter]

10 May 2024

PONE-D-23-37663R2 

PLOS ONE

Dear Dr. Burt, 

I'm pleased to inform you that your manuscript has been deemed suitable for publication in PLOS ONE. Congratulations! Your manuscript is now being handed over to our production team.

Kind regards, 

on behalf of

Dr. Travis Longcore 

Academic Editor

PLOS ONE